# Large-scale digital forensic investigation for Windows registry on Apache Spark

**Jun-Ha Lee[1], Hyuk-Yoon Kwon**[2]*

**1** Department of Industrial Engineering, Seoul National University of Science and Technology, Nowon-Gu, Seoul, Korea, **2** Department of Industrial Engineering, Graduate School of Data Science, and the Research Center for Electrical and Information Technology, Seoul National University of Science and Technology, Nowon-Gu, Seoul, Korea

* hyukyoon.kwon@seoultech.ac.kr

**Data Availability Statement:** All the data files are available from Mendeley Data (doi: 10.17632/ghkms3cgbg.1).

**Funding:** This work was supported in part by the National Research Foundation of Korea(NRF) grant funded by the Korea government(MSIT) (No.

## Abstract

In this study, we investigate large-scale digital forensic investigation on Apache Spark using a Windows registry. Because the Windows registry depends on the system on which it operates, the existing forensic methods on the Windows registry have been targeted on the Windows registry in a single system. However, it is a critical issue to analyze large-scale registry data collected from several Windows systems because it allows us to detect suspiciously changed data by comparing the Windows registry in multiple systems. To this end, we devise distributed algorithms to analyze large-scale registry data collected from multiple Windows systems on the Apache Spark framework. First, we define three main scenarios in which we classify the existing registry forensic studies into them. Second, we propose an algorithm to load the Windows registry into the Hadoop distributed file system (HDFS) for subsequent forensics. Third, we propose a distributed algorithm for each defined forensic scenario using Apache Spark operations. Through extensive experiments using eight nodes in an actual distributed environment, we demonstrate that the proposed method can perform forensics efficiently on large-scale registry data. Specifically, we perform forensics on 1.52 GB of Windows registry data collected from four computers and show that the proposed algorithms can reduce the processing time by up to approximately 3.31 times, as we increase the number of CPUs from 1 to 8 and the number of worker nodes from 2 to 8. Because the distributed algorithms on Apache Spark require the inherent network and MapReduce overheads, this improvement of the processing performance verifies the efficiency and scalability of the proposed algorithms.

## 1 Introduction

In this study, we investigate large-scale digital forensic investigation on Apache Spark using a Windows registry. The Windows registry is a tree-structured database that stores necessary information for the Windows operating system and the installed programs such as version information, configurations, and the referencing file locations [1]. In the Windows registry, the registry key and its corresponding registry value are stored in the form of a key-value pair.

2021R1F1A1064050), and in part by the Basic Science Research Program through the National Research Foundation of Korea(NRF) funded by the Ministry of Education (No. 2019R1A6A1A03032119). The funders had no role in study design, data collection and analysis, decision to publish, or preparation of the manuscript.

Fig 1 shows the structure of the Windows registry. Each registry key can recursively have subkeys.

The Windows registry stores critical information, including user accounts and program locations executed, when the system is booting [2, 3]. This information can be abused for cyberattacks such as dll hijacking, malware persistence, and privilege escalation by manipulating the stored information. This implies that the data stored in the Windows registry are critical evidence for digital forensics to effectively detect cyberattacks. As a result, forensics on the Windows registry is one of the representative forensic types in Windows systems [4–6].

In this study, we devised distributed algorithms to analyze large-scale registry data collected from a number of Windows systems on the Apache Spark framework. Fig 2 shows the overall proposed framework compared to the existing forensic approach on the Windows registry. The existing approach analyzes Windows registry targeting on a single Windows system. However, it can perform forensic analysis only for a single registry and has a limitation to identify malicious entries in a registry based on the comparison with other multiple Windows registry repositories. On the other hand, in our framework, we extract the Windows registry from several Windows systems and transform and load them into the Hadoop distributed file system (HDFS) on a Hadoop cluster. Then, we perform forensic analysis on large-scale data files stored on HDFS. Therefore, we can compare multiple Windows registry repositories in a scalable way and identify malicious registry entries based on them.

The contributions of the paper are summarized as follow:

1. In this study, we manage and analyze large-scale Windows registry collected from multiple systems. For this purpose, we present an algorithm that transforms and loads the Windows registry collected from multiple systems to HDFS using Apache Spark operations. This allows forensics on large-scale Windows registry on a Hadoop cluster, which provides scalability for storing large-scale Windows registry.

2. We propose distributed algorithms using Apache Spark operations for forensics on the Windows registry stored on HDFS. We define three main scenarios by classifying the existing registry forensic methods into the scenarios and propose a distributed algorithm for each scenario. The proposed algorithms allow us to perform forensics faster than that in a single machine by processing it in parallel on Apache Spark using multiple nodes in a Hadoop cluster.

3. The proposed method was used to conduct forensics on 1.52 GB of Windows registry data collected from four Windows systems. Consequently, we show that the proposed method can reduce the processing time by up to approximately 3.31 times as we increase the number of CPUs from 1 to 8 and the number of worker nodes from 1 to 8 because of the proposed algorithms leveraging distributed and parallel processing effectively, thereby validating the efficiency and scalability of the proposed algorithms.

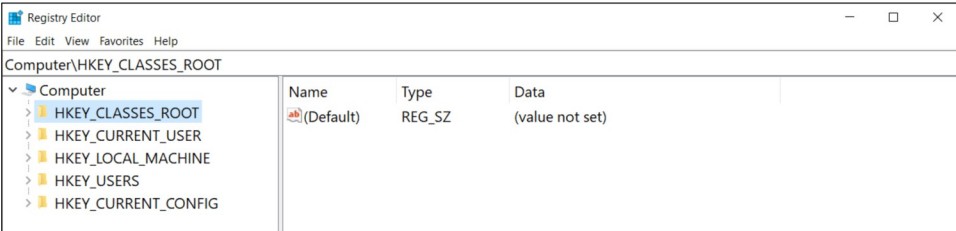

**Fig 1. The structure of Windows registry.**

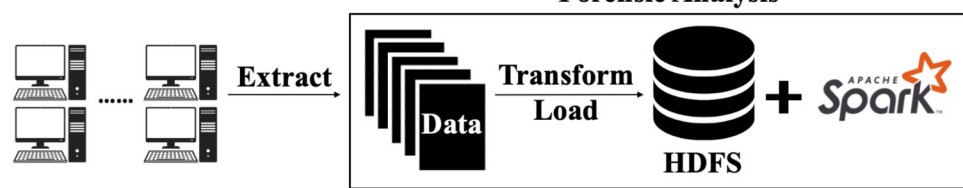

**Forensic Analysis**

(**a**) The existing forensic strategy for a single system.

(**b**) The proposed forensic strategy for multiple Windows systems.

**Fig 2. The overall framework of the proposed forensic analysis on Windows registry.** (**a**) The existing forensic strategy for a single system. (**b**) The proposed forensic strategy for multiple Windows systems.

The remainder of this paper is organized as follows. In Section 3, we extensively survey and review existing forensic analysis techniques. In Section 2, we explain the Windows registry, MapReduce, and Apache Spark as the background of the paper. In Section 4, we define three forensic scenarios and propose an algorithm using Apache Spark operations for each scenario. In Section 5, we present the experimental results. Finally, we present our conclusions in Section 6.

## 2 Background

### 2.1 Windows registry

To extract the Windows registry from multiple Windows systems, we need a consistent approach to extract it from each different version of Windows system including servers, PCs, and mobile versions. In this study, we extract the registry data from a target Windows system using **Regedit** [7], which is the built-in registry editor supported in any version of Windows systems. Fig 3 shows an example of the exported file. We note that we can apply the existing forensic methods for Windows registry into the exported text file because all the information in the original registry is maintained in the text file, including the hierarchies of registry keys, i.e., the relationship between registry keys and their subkeys [2]. Here, we can collect all the registry keys and values stored in a system only by specifying the root key of the registry.

### 2.2 MapReduce and Apache Spark

MapReduce is a framework developed by Google for providing distributed and parallel computing of large-scale datasets based on a cluster of multiple nodes connected on the network [8]. The MapReduce framework consists of two stages: 1) **map** stage, which partitions the entire dataset into multiple data chunks and assigns each data chunk to one node, and 2) **reduce** stage, which aggregates the sub-results obtained from all the involved nodes as a final result.

```
[HKEY_CURRENT_CONFIG]

[HKEY_CURRENT_CONFIG\Software]

[HKEY_CURRENT_CONFIG\Software\Fonts]
"LogPixels"=dword:00000060

[HKEY_CURRENT_CONFIG\System]

[HKEY_CURRENT_CONFIG\System\CurrentControlSet]

[HKEY_CURRENT_CONFIG\System\CurrentControlSet\Control]

[HKEY_CURRENT_CONFIG\System\CurrentControlSet\Control\Print]

[HKEY_CURRENT_CONFIG\System\CurrentControlSet\Control\Print\Printers]

[HKEY_CURRENT_CONFIG\System\CurrentControlSet\SERVICES]

[HKEY_CURRENT_CONFIG\System\CurrentControlSet\SERVICES\TSDDD]

[HKEY_CURRENT_CONFIG\System\CurrentControlSet\SERVICES\TSDDD\DEVICE0]
"Attach.ToDesktop"=dword:00000001
```

**Fig 3. An example of the registry data exported by Regedit.**

Apache Spark is an open-source software platform that supports the MapReduce framework [9, 10]. Apache Spark works with underlying systems for distributed environments, i.e., HDFS [11], for storing and managing large-scale data files and YARN for managing resources in a distributed environment. Because it can manage large-scale data that cannot be stored in a single computer and provide parallel processing using multiple nodes in distributed environments, there have been many research efforts to apply the traditional problem in a single computer to a distributed environment using Apache Spark platform [12, 13]. Hsu et al. [14] reduced the time of processing and mining the tweet data that can be used as evidence for drug side effects by up to 2.5 times by partitioning the dataset into two nodes with 12 cores and processing them based on Apache Spark. Harine et al. [15] reduced the processing time of machine learning algorithms that predict molecules affecting proteins associated with diseases by up to 14.19 times based on Apache Spark using 20 nodes.

## 3 Related work

### 3.1 Windows forensics

We can classify existing studies on Windows forensics as follows: 1) registry forensics, 2) memory forensics, and 3) application forensics. Registry forensics investigate the data stored in the Windows registry to detect evidence of cyberattacks. Venčkauskas et al. [16] captured the entire registry and examined every stage of the installation, execution, and removal of a certain program. Wong et al. [17] and Farmer et al. [18] investigated recently updated registry keys using the keytime.exe software. Alghafli et al. [19] classified registry keys into categories such as systems, applications, and networks and examined the registry values contained in each category. Verma et al. [20] collected the whole registry using an open source tool, Regshot, and compared two registry repositories captured at different time points to detect newly installed malicious software. Casey et al. [4] performed forensic analysis on a certain registry key, 'HKCU\Software\RetinaxStudios', to trace a mobile surveillance program, MobileSpy. Rehualt

[21] classified the registry data in the mobile device based on specific suspicious directories such as 'settings\default\user.hv' and 'settings\system.hv' and demonstrated that forensics for Windows mobile phones can be performed the same as in Windows PCs. Roy and Jain [22] examined registry keys created when the PC recognizes USB, e.g., 'HKLM\System\Control-Set00x\Enum\USBSTOR\device_class\device_unique_id'. Klaver [23] performed forensic analysis not only for active data but also for data deleted on Windows mobile devices using various recovery techniques such as chip extraction, using JTAG, and a boot loader.

Memory forensics investigate data residing in memory to detect malicious actions of running programs. Ruff et al. [5] employed various techniques for capturing real-time data in memory, such as CrashDump and Snapshot, and verified their effectiveness through practical examples. Schuster et al. [24] proposed pool allocation mechanisms to collect volatile data on the current and past processes and performed forensic analysis on them. Canlar et al. [25] proposed a real-time data acquisition method from both random access memory (RAM) and electronic enabling programmable read-only memory (EEPROM) on a Windows mobile device.

Application forensics investigate the results operated by certain applications to detect malicious actions. Gianni and Solinas [6] performed real-time forensic analysis on two different versions of Windows, i.e., Windows XP and Windows 7, targeting common applications such as Skype, Google Talk, and Internet Explorer. Yang et al. [26] analyzed the artifacts that remained after the instant messaging services of Facebook and Skype were run on the Windows operating system. Murphey [27] proposed a method for extracting complete log data for forensics, including not only the normal Windows event log data but also the log data that cannot be currently accessed by repairing and recovering them. Chang et al. [28] examined artifacts remaining after various actions such as installation, uninstallation, log-in, chatting, and file transferring occurred by the LINE application on Windows 10.

Other existing studies on Windows forensics are as follows. Ahmadi et al. [29] minimized the time required for imaging in forensics, which creates a duplicate of the media, by targeting only necessary information such as system logs, Windows registry, and recycle bin instead of the entire data. Yang et al. [30] proposed an efficient forensic method targeting a cloud storage service such as CloudMe.

As described, the existing Windows forensic studies focused on the registry entries extracted from a single system. They usually focused on target Windows registries where it has been known that suspicious information is stored. On the other hand, in this study, we propose a scalable forensic analysis of Windows registry by introducing Spark-based forensic framework to compare the entire Windows registry repositories collected from multiple systems, focusing on the differential registry entries.

## 3.2 Big data forensics

Zawoad and Hasan [31] proposed a conceptual model for big data forensics based on HDFS and cloud architecture and reduced the entire forensic time by eliminating redundant data from incoming streaming data. Adedayo et al. [32] presented new challenges and opportunities for forensic analysis of large-scale data such as identification, collection, organization, preservation, and presentation. Thaneker et al. [33] proposed a forensic framework on large-scale data that uses Hadoop for storing and managing forensic evidence, and used Autopsy, an open-source forensic tool, for file carving, data carving, and keyword searching.

Qi et al. [34] dealt with forensic analysis of large-scale data by considering four types of NoSQL databases as an alternative to RDBMS: 1) key-value databases, 2) document databases, 3) column-family databases, and 4) graph databases. Then, they evaluated the processing

performance of the forensic analysis using two representative NoSQL databases, i.e., MongoDB and Riak, showing that Riak performs better than MongoDB.

In summary, to the best of our knowledge, there have been no research efforts that focus on forensic analysis on the Windows registry in distributed nodes to deal with large-scale data. Instead, as described, there have been previous studies on large-scale data except for the Windows registry. Furthermore, most studies have focused on the concept of big data forensics or the overall framework rather than specific algorithms.

### 3.3 Digital forensics using Apache Spark and MapReduce

There have been few existing studies on forensic analysis using Apache Spark. Hemdan et al. [12] examined large-scale log data collected from the cloud service's web server using Apache Spark to reconstruct cyber crimes. Gonzales et al. [13] proposed a forensic framework based on distributed computing platforms such as Apache Spark and Kafka to collect and detect forensic evidence stored on hard drives. Then, they demonstrated that the performance of the distributed computing platform was substantially faster than the standalone version of Autospy. Chhabra et al. [35] proposed a forensic framework that uses MapReduce to analyze dynamic traffic features from large-scale traffic data generated from IoT devices such as Raspberry Pi for malicious traffic detection. Guarino [36] introduced how big data analysis techniques and algorithms such as MapReduce and decision trees can be adapted to each step of digital forensics: identification, collection, acquisition, preservation, analysis, and reporting.

These existing studies have also performed digital forensics through distributed frameworks as the same as our study. However, they have not considered the forensic analysis of Windows registry based on distributed frameworks. In this study, we present Apache Spark-based forensic analysis for Windows registry and propose a scalable analysis methodology for dealing with multiple Windows registry repositories for the first time.

## 4 The proposed method

### 4.1 Forensic scenarios

We define three forensic scenarios through an investigation of the existing studies on the Windows registry forensic and map existing studies into these scenarios as follows:

**Scenario 1**. Forensic for target registry keys.

- **Case 1**. To trace MobileSpy, one of the mobile surveillance programs, performing forensic analysis for a certain registry key, 'HKCU\Software\RetianxStudios', which is created when the program is installed [4]

- **Case 2**. Performing forensic analysis on system-relevant registry keys(e.g., 'HKLM\Software \Apps', information of the installed softwares, and 'HKLM \System\Uptime', time of the last system booting) and user-relevant registry keys(e.g., 'HKCU\ControlPanel\Owner', owner information, and 'HKCU \Software\Microsoft\ActiveSync', smartphone UID used when syncing with a computer) [21]

- **Case 3**. Identifying registry keys created when the computer systems recognize the USB(e.g., 'HKLM\System \ControlSet00x\Enum\USBTOR \<device_unique_id>') [22]

- **Case 4**. Classifying the registry keys into systems, applications, and networks, and performing forensic analysis on each category [19]

- **Case 5**. Classifying the registry keys selected for forensics into three types: hardware, software, and network, and performing forensic analysis on those registry keys [37]

**Scenario 2**. Forensic for registry keys and values containing target keywords.

- **Case 1**. According to an APT report by Kaspersky laboratory [38], 'fileless attacks against enterprise networks', the following strings were contained in the registry: 'powershell.exe -nop -w hidden -e', '10.10.1.11\8080', and '10.10.1.11\4444'

- **Case 2**. According to an APT report by Kaspersky Lab [39], 'Lazarus Under the Hood,' the following two Windows commands were contained in the registry: 'C:\Users\%user%\App-Data\Roaming\IMEKLMG.exe -s' and '%self_path%' -k %param1% %param2%'

- **Case 3**. According to an APT report analyzed by ElevenPaths [40], 'APTualizator: the malware patching Windows,' strings such as 'S02', 'S03', 'S04', 'S05', and 'S0' were contained in found from the registry

  **Scenario 3**. Comparison between the entire registry repositories

- **Case 1**. Comparing the entire registry repository in Windows 10 and that in Windows 8.1 to identify the possible forensic targets that are newly introduced in Windows 10 [41]

- **Case 2**. Capturing the entire registry repository for each step when certain programs are installed, executed, and removed, and comparing the captured registry sets [16]

- **Case 3**. Capturing the entire registry repository using Regshot, an open-source tool that captures the entire registry, and comparing two registry repositories captured at different time points [20]

## 4.2 Registry data processing

The Windows registry is formed as a tree-based structure consisting of key-value pair entries. However, the data exported from the Windows registry are stored in a text file, as shown in Fig 3. In this section, we present algorithms for manipulating the exported registry data in the form of a tree structure to maintain the original hierarchy of the registry keys.

**4.2.1 Converting the registry entry to nested key-value data.** We convert the text data exported from the Windows registry into the form of key-value data so we can access each level of the registry key in the tree structure. For this purpose, we encapsulate each level of the subkey as the value of the key-value data at a nested level. For example, a registry entry, 'HKCC\Software\Fonts \Logpixels = dword:00000060', is converted into the following nested key-value data, '{HKCC: {Software: {Fonts: {Logpixels: dword:00000060}}}}'.

Algorithm 1 shows *ConvertRegEntryToNestedKeyValue()*, which converts a registry entry into a form of nested key-value data. It receives *RegEntry*, which is a line of the text data exported from the Windows registry, as the input and returns *RegNestedKeyValue*, which is a nested key-value data converted from *RegEntry*. In Lines $1 \sim 3$, the algorithm separates the registry key and the registry value by '='. Then, because each level of registry subkey is distinguished by '\', the algorithm splits the entire registry key path by '\' and stores them into *keys* as an array while storing the registry value into *value*. In Lines $4 \sim 8$, we construct *nestedValue* by appending each level of key path into the form of key-value pairs. This process is repeated until all subkeys are appended. The final result stored in *RegNestedKeyValue* becomes the nested key-value data for an input *RegEntry*.

**Algorithm 1**: *ConvertRegEntryToNestedKeyValue()*

```
Input: RegEntry
Output: RegNestedKeyValue
1: keys = RegEntry.split('=')[0].split('\')
```

```
2: value = RegEntry.split('=')[1]
3: reg = keys + value
4: while type(reg[0]) != dict:
5:   regValue = reg.pop()
6:   regKey = reg.pop()
7:   nestedValue = { regKey: regValue }
8:   RegNestedKeyValue.append(nestedValue)
```

Fig 4 shows an example of the process of converting 'HKCC\Software\Fonts \Logpixels = dword:00000060', an actual line of Windows registry data in Fig 3, into a nested key-value data according to the *ConvertRegEntryToNestedKeyValue()* algorithm. As can be observed in Fig 4, 'LogPixels = dword:00000060' is recognized as the first key-value pair. Then, it becomes the value of the the key 'Fonts' for the next level. This process is repeated until we meet a registry hive, HKCC.

**4.2.2 Merging registry entries based on the common registry key path.** In this section, we present a method to merge registry entries based on the same common registry key path and convert it into a list for a single common key path. This method scans two adjacent registry entries for the entire registry repository and merges based on the common registry. By applying this method to the flattened text file extracted from the Windows registry, we can construct tree-structured data from the text file. For example, two nested registry entries, '{HKU: {Control Panel:' {Mouse: {Mouse Speed: 1}} '}}' and '{HKU: {Control Panel:' {International: {Geo: {Name:KR}}} '}}', are merged into a list of '{Mouse: {Mouse Speed: 1}}' and '{International: {Geo: {Name:KR}}}' for the common key path, '{HKU: {Control Panel}}'.

Algorithm 2 shows *MergeRegNestedEntries()* that merges registry entries based on the common registry key path. It receives two nested registry entries, $RNE_1$ and $RNE_2$, which are obtained by *ConverteRegEntryToNestedKeyValue()*, as the input and returns the registry entries that are merged by the common registry key path, $RNE_{1+2}$. In Lines $2 \sim 8$, the algorithm finds the common registry key path of $RNE_1$ and $RNE_2$ by comparing each level of $RNE_1$ and $RNE_2$ from the root key until the subkeys are different. The common registry key path is stored in *keys*. In Line 9, the remaining registry keys and values in $RNE_1$ and $RNE_2$ are stored in a list for the result value. In Lines $10 \sim 12$, it constructs the common path and the list as one structure, $RNE_{1+2}$.

**Algorithm 2**: *MergeRegNestedEntries()*

```
Input: RNE₁, RNE₂
Output: RNE₁₊₂
1: keys = []
2: while True:
3:   if RNE₁.keys() == RNE₂.keys():
4:     keys.append(RNE₁.keys())
5:     RNE₁ = RNE₁.values()
6:     RNE₂ = RNE₂.values()
7:   else:
8:     break
9: ListRegEntries = RNE₁.values().update(RNE₂.values())
10: while keys:
11:   commonPath = { keys.pop(): commonPath }
12: RNE₁₊₂ = { commonPath: ListRegEntries }
```

Fig 5 shows an example of the process of merging three actual registry entries according to the *MergeRegNestedEntries()* algorithm. Fig 5(a) shows an example of three registry entries. Fig 5(b) shows a process of merging the first two registry entries, $RNE_1$ and $RNE_2$, based on common registry key path. '{HKU:{Control Panel:{}}}' becomes the common registry key path of $RNE_1$ and $RNE_2$. The remaining subkeys and registry values, '{Mouse:{MouseSpeed:1}, International:{Geo:{Name:KR}}}' become a list for the result value. Fig 5(c) shows the subsequent

**Fig 4. An example of the actual process of converting a registry entry into the nested key-value data.**

(**a**) The example of three actual registry entries.

(**b**) The result of the first *MergeRegNestedEntries*().

(**c**) The result of the second *MergeRegNestedEntries*().

**Fig 5. An example of the actual process of merging three registry entries based on the common registry path.** (**a**) The example of three actual registry entries. (**b**) The result of the first *MergeRegNestedEntries*(). (**c**) The result of the second *MergeRegNestedEntries*().

step to the result in Fig 5(b). It shows a process of merging $RNE_1$ + 2, with another registry entry, $RNE_3$. '{HKU:{Control Panel:{}}}' becomes the common registry key path and the remaining subkey and registry value in $RNE_3$, '{Desktop:{Status:True}}', is added to $RNE_1$ + 2. By applying *MergeRegNestedEntries()* into adjacent registry entries continuously, we can finally construct a tree structure.

**4.2.3 Comparing registry entries.** It is worthwhile to detect the differences between two entire Windows registry data targeting the following two cases: 1) a certain registry key and the associated registry value exist only in one Windows registry or 2) different values for the same registry key, which is crucial evidence for forensics. In this section, we present an algorithm to compare one registry entry from a registry repository with another registry repository. This will be used as the basic function for the algorithm to compare two entire registry repositories using Apache Spark operations in Section 4.4.3.

Algorithm 3 shows *ComparingRegEntries()*, which compares a registry entry from a registry repository with the entire registry entries from another registry repository. Here, a registry entry exported from one registry repository, *RegNestedEntry*, is compared with the entire registry entries exported from another registry repository, *RegRepository*. *RegNestedEntry* is nested key-value data converted from a registry entry by calling *ConvertRegEntryToNestedKey-Value()*. *RegRepository* is the entire Windows registry data where all the registry entries are converted into nested key-value data and then transformed into a single tree structure by calling *MergeRegNestedEntries*() into adjacent registry entries.

The algorithm compares *RegNestedEntry* with each entry in the *RegRepository* in the while loop. The algorithm identifies two suspicious cases. First, the registry key of *RegNestedEntry* does not exist in the *RegRepository*. Second, the registry key of *RegNestedEntry* exists in the *RegRepository*, but their registry values are different. In Lines 5 ∼ 11, it checks the registry keys between two entries. Because all the subkeys in each level of the registry repository are maintained in *RegRepository*, in Line 6, we can easily check if each level of the registry subkey in *RegNestedEntry* exists in *RegRepository*. In Lines 7 ∼ 8, we inspect the next level of registry subkey only if the previous registry subkeys are the same. If the algorithm finds the different subkeys between them, it appends *RegNestedEntry* into *CompResult*. If the algorithm reaches Line 13, this means that the registry key of *RegNestedEntry* exists in *RegRepository*. In Lines 13 ∼ 18, the algorithm checks if their registry values are different. If the value of *RegNestedEntry* is different from that of the corresponding entry in the *RegRepository*, it appends *RegNestedEntry* into *CompResult*.

**Algorithm 3**: *ComparingRegEntries*()

```
Input: RegNestedEntry, RegRepository
Output: CompResult
1: regTemp = RegNestedEntry
2: while True:
3:   // forensic for Registry Key
4:   key = regTemp.keys()[0]
5:   if key:
6:     if key in RegReporitory.keys():
7:       RegRepository = RegRepository[key]
8:       regTemp = regTemp[key]
9:     else:
10:       CompResult = RegNestedEntry
11:       break
12:   // forensic for Registry Value
13:   else:
14:     value = regTemp
15:     if RegRepository != value:
```

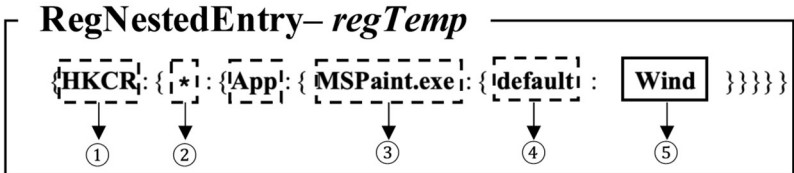

**Fig 6. An example of the actual process of comparing two registry entries.**

```
16:        CompResult = RegNestedEntry
17:     else:
18:        break
```

Fig 6 shows an example of the process of comparing two registry entries according to the *ComparingRegEntries()* algorithm. First, the key of *RegNestedEntry*, 'HKCR', is included in a set of keys in *RegRepository*, '[HKCR]'. Similarly, '*', 'App', 'MSPaint.exe', and 'default' are included in a set of keys in *RegRepository*. Because we know that the registry key of *RegNestedEntry* exists in *RegRepository*, we compare their values. Because 'Wind' are not included in a set of values in *RegRepository*, *RegNestedEntry* becomes the result.

## 4.3 Loading registry data into Hadoop distributed file system

To analyze the Windows registry data based on the Apache Spark framework, we need to load Windows registry data into HDFS, which is an underlying storage for Apache Spark. In this section, we present an algorithm to process it using Apache Spark operations. Algorithm 4 shows the algorithm for loading the Windows registry into HDFS. The inputs of the algorithm are the registry data exported from **Regedit**, *ExportedData*, and the desired number of partitions, *nPartitions*. The detailed steps of Algorithm 4 are as follows. In Line 2, the algorithm converts the entire Windows registry data into a form of RDD that can be accessed on a distributed cluster environment using **parallelize()**. In Line 3, it repartitions the entire RDD by *nPartitions* using **repartition()**. This operation can be commonly used to control the actual number of partitions in the experiments. In Line 4, the algorithm separates the Windows registry data by a new line character because **Regedit** exports each entry of the registry data into a single line using **flatMap()**. In Line 5, it converts each registry entry into the form of nested key-value data using **map()** by calling a function *ConvertRegEntrytoNestedKeyValue()*. In Line 6, it merges two registry entries using **reduce()** by calling a function *MergeRegNestedEntries()*. Finally, the result RDD, *regRDD*, becomes a single tree structure stored on HDFS.

**Algorithm 4**: Loading Windows Registry into HDFS

```
Input: ExportedData, nPartitions
Output: regRDD
1: sc = spark.SparkContext()
2: rdd = sc.parallelize(ExportedData)
3: rdd.repartition(nPartitions)
4: rdd.flatMap(lambda x: x.split('\n'))
5: rdd.map(lambda x: ConvertRegEntryToNestedKeyValue(x))
6: regRDD = rdd.reduce(lambda x,y: MergeRegNestedEntries(x,y))
```

Fig 7 shows an example of the process of loading Windows registry data into HDFS according to Algorithm 4. It describes the result of each step from Lines 4∼6 for four registry entries. In Line 4, we separate four registry entries. In Line 5, we convert each of them into the form of nested key-value data. In Line 6, we merge them based on the common registry key path. As a result, we obtain a tree structure where '{HKCC:{System:{CurrentControlSet:{}}}}' is the common path of the tree and 'Control:{}' and 'SERVICES:{}' are the next level of the tree and 'Print:{}' and 'TSDDD:{}' are the leaf level of the tree.

## 4.4 Forensics analysis on Windows registry on Apache Spark

In this section, we propose algorithms using Apache Spark operations to perform forensic analysis on the Windows registry for the three scenarios defined in Section 4.2.1.

**4.4.1 Forensic for target registry keys.** Algorithm 5 shows *ForensicForTargetRegKey()* that retrieves the registry entry for a target registry key. The inputs of the algorithm are the registry data exported from **Regedit**, *ExportedData*, the desired number of partitions, *nPartitions*, and a target registry key for forensics, *TargetRegKey*. The detailed steps of Algorithm 5 are as follows. Here, a sequence of operations in Lines 2∼5, i.e., **parallelize()**, **repartition()**, **flatMap** (), and **map()**, which transforms each registry entry into an RDD form of the nested key-value data, are the same as in Algorithm 4. In Line 6, the algorithm finds the registry entry whose registry key path exactly equals *TargetRegKey*.

**Algorithm 5**: *ForensicForTargetRegKey()*

```
Input: ExportedData, nPartitions, TargetRegKey
Output: TargetRegEntry
1: sc = spark.SparkContext()
2: rdd = sc.parallelize(ExportedData)
3: rdd.repartition(nPartitions)
```

| **Line 4** | HKCC\System\CurrentControlSet\Control |
| | HKCC\System\CurrentControlSet\Control\Print |
| | HKCC\System\CurrentControlSet\SERVICES |
| | HKCC\System\CurrentControlSet\SERVICES\TSDDD |

↓

**Line 5**  {HKCC : {System : {CurrentControlSet : {Control : {}}}}}}
{HKCC : {System : {CurrentControlSet : {Control : { Print : {}}}}}}}
{HKCC : {System : {CurrentControlSet : {SERVCIES : {}}}}}}
{HKCC : {System : {CurrentControlSet : {SERVCIES : {TSDDD : {}}}}}}}}

↓

**Line 6**  {HKCC : {System : {CurrentControlSet : { Control       : { Print : {}},

SERVICES  : { TSDDD : {}}}}}}

⌞___Common Path(Level 1)___⌟  ⌞_Level 2_⌟  ⌞_Level 3_⌟

**Fig 7. An example of the actual process of loading Windows registry data into HDFS.**

```
4: rdd.flatMap(lambda x: x.split('\n'))
5: rdd.map(lambda x: ConvertRegEntrytoNestedKeyValue(x))
6: TargetRegEntry = rdd.filter(lambda x: x == TargetRegKey)
```

Fig 8 shows an example of the process of forensic for a target registry key according to Algorithm 5. Here, a registry key path, 'HKCR\*\App\MSPaint.exe \content', is given as *TargetRegKey*. In Lines 4∼5, we process the same steps as in Algorithm 4. In Line 6, we find the registry entry whose registry key is the same as *TargetRegKey*.

**4.4.2 Forensic for registry entries containing keywords.** Algorithm 6 shows *ForensicRegEntriesUsingKeywords()*, which performs forensic analysis for registry entries containing

| **Line 4** | HKCR\*\App\MSPaint.exe "default"="Window" |
| | HKCR\*\App\MSPaint.exe "content"="image" |
| | HKCR\*\App\WordPad.exe "default"="Window" |
| | HKCR\*\App\WordPad.exe "content"="text" |

↓

**Line 5**  {HKCR:{*:{App:{MSPaint.exe:{default:Window}}}}}
{HKCR:{*:{App:{MSPaint.exe:{content:image}}}}}
{HKCR:{*:{App:{WordPad.exe:{default:Window}}}}}
{HKCR:{*:{App:{WordPad.exe:{content:text}}}}}

↓

**Line 6**  {HKCR:{*:{App:{MSPaint.exe:{content:image}}}}}

**Fig 8. An example of the actual process of forensic for a target registry key.**

keywords. The inputs of the algorithm are the registry data exported from **Regedit**, *Exported-Data*, the desired number of partitions, *nPartitions*, and a target keyword to find, *TargetKeyword*. The detailed steps of Algorithm 6 are as follows. Here, a sequence of operations in Lines 2∼5, i.e., **parallelize**(), **repartition**(), **flatMap**(), and **map**() are the same as in Algorithm 4 and Algorithm 5. In Line 6, we find the registry entries whose keys or values contain the given *TargetKeyword* using **filter**(). In Line 7, we aggregate all the found registry entries and merge them into a form of the tree-structured data using **reduce**().

**Algorithm 6**: *ForensicRegEntriesUsingKeywords*()

```
Input: ExportedData, nPartitions, TargetKeyword
Output: RegNestedEntry
1: sc = spark.SparkContext()
2: rdd = sc.parallelize(ExportedData)
3: rdd.repartition(nPartitions)
4: rdd.flatMap(lambda x: x.split('\n'))
5: rdd.map(lambda x: ConvertRegEntryToNestedKeyValue(x))
6: rdd.filter(lambda x: TargetKeyword in x)
7: RegNestedEntry = rdd.reduce(lambda x,y: MergeRegNestedEntries(x,
y))
```

Fig 9 shows an example of the process of forensics on registry entries using keywords according to Algorithm 6. Here, a keyword 'MS' is given as *TargetKeyword*. In Lines 4∼5, we process the same steps as in Algorithm 5. In Line 6, we find the registry entries whose registry keys or registry values contain 'MS'. Here, the first result contains 'MS' in the registry key; the second result contains it in both registry key and value; the third result contains it in the registry value. In Line 7, we merge them as the nested key-value data.

**4.4.3 Comparing the entire registry repositories.** Algorithm 7 shows *CompareRegRepositories()* that compares the entire two Windows registry repositories and finds the differences by extending the *ComparingRegEntries()* algorithm in Section 4.2.3, which compares a registry entry from one registry repository with another registry repository. The inputs of the algorithm are two registry repositories to compare, i.e., *RegRepo$_1$* and *RegRepo$_2$*, and the desired number of partitions, *nPartitions*. The detailed steps of Algorithm 7 are as follows. In Lines 2∼6, the algorithm converts the entire registry repository, *RegRepo$_1$*, into the nested key-value data, *NestedRegRepo$_1$*. Then, in Lines 7∼10, it compares the entire two Windows registry repositories using **map**() with calling of *ComparingRegEntries()*. In Line 11, the algorithm aggregates all the result registry entries of *RegRepo$_1$* that are different from the corresponding registry entry in *RegRepo$_2$* using **reduce()** by calling *MergeRegNestedEntries()*.

**Algorithm 7**: *CompareRegRepositories*()

```
Input: RegRepo₁, RegRepo₂, nPartitions
Output: DifferentRegEntries
1: sc = spark.SparkContext()
2: rdd₁ = sc.parallelize(RegRepo₁)
3: rdd₁.repartition(nPartitions)
4: rdd₁.flatMap(lambda x: x.split('\n'))
5: rdd₁.map(lambda x: ConvertRegEntryToNestedKeyValue(x))
6: NestedRegRepo₁ = rdd₁.reduce(lambda x,y: MergeRegNestedEntries(x,
y))
7: rdd₂ = sc.parallelize(RegRepo₂)
8: rdd₂.repartition(nPartitions)
9: rdd₂.flatMap(lambda x: x.split('\n'))
10: rdd₂.map(lambda RegEntry: ComparingRegRepositories(NestedRe-
gRepo₁, ConvertRegEntryToNestedKeyValue(RegEntry))
11: DifferentRegEntries = rdd₂.reduce(lambda x,y: MergeRegNestedEn-
tries(x,y))
```

| Line 4 | HKCR\\*\App\MSPaint.exe "default"="Window"<br>HKCR\\*\App\MSExcel.exe "default"="MSoffice"<br>HKCR\\*\App\WordPad.exe "default"="Window"<br>HKCR\\*\App\WinWord.exe "default"="MSoffice" |
|---|---|
| Line 5 | {HKCR:{*:{App:{MSPaint.exe:{default:Window}}}}}<br>{HKCR:{*:{App:{MSExcel.exe:{default:MSoffice}}}}}<br>{HKCR:{*:{App:{WordPad.exe:{default:Window}}}}}<br>{HKCR:{*:{App:{WinWord.exe:{default:MSoffice}}}}} |
| Line 6 | {HKCR:{*:{App:{*MS*Paint.exe:{default:Window}}}}}<br>{HKCR:{*:{App:{*MS*Excel.exe:{default:*MS*office}}}}}<br>{HKCR:{*:{App:{WinWord.exe:{default:*MS*office}}}}} |
| Line 7 | {HKCR:{*:{App:{*MS*Paint.exe:{default:Window},<br>{*MS*Excel.exe:{default:*MS*office},<br>{WinWord.exe:{default:*MS*office}}}}} |

**Fig 9. An example of the actual process of forensic on registry entries using keywords.**

Fig 10 shows an example of the process of comparing the entire two registry repositories according to Algorithm 7. In Lines $2 \sim 6$, we construct a tree-structure for $RegRepo_1$. In Lines $7 \sim 10$, we compare each registry entry in $RegRepo_2$ with $RegRepo_1$ and find the differences. In Line 11, we merge them based on the common registry key path.

## 5 Performance evaluation

### 5.1 Experimental environments and data sets

In this section, we measure the processing time of the proposed distributed algorithms using the scenario. Here, we use actual distributed environments where eight worker nodes are configured based on Apache Spark to show the effectiveness of the algorithms. The effects of distributed processing on Apache Spark are achieved by the following three aspects: 1) providing scalable storage by loading the entire dataset into the cluster over multiple nodes, 2) parallel processing using multiple nodes, and 3) parallel processing within a node using multi-threads with multiple cores. For the first effect, we measure the processing time of the algorithm for loading registry data into HDFS proposed in Section 4.3 using up to eight nodes. For the second effect, we measure the processing time of the three scenarios by changing the number of nodes from 1 to 8. For the third effect, we measure the processing time of the three scenarios by changing the number of CPUs from 1 to 8 on each node.

**Fig 10. An example of the actual process of comparing the entire two registry repositories.**

In this paper, we use Google Cloud Platform [42] to build an actual distributed environment where Apache Hadoop 2.10.0 [43] and Apache Spark 2.4.7 [44] are installed. A cluster consists of one master node that manages the overall operations and up to eight slave nodes. Each node is equipped with 2.0 GHz of 8 vCPUs, 3.75GB of memory, and 128GB of disk size. We collected actual Windows registry data from four different systems running Windows operating systems.

Table 1 lists the details of the collected Windows registry data. We can extract all the registry entries for a system into a text file with a Windows built-in command "regedit /E" [2]. Hence, we can easily collect the registry entries from all the Windows-installed systems including servers, desktops, and mobile devices, regardless of the Windows versions. The size of Windows registry files collected from Windows 10 depends on the system, i.e., 240MB, 352MB, 413MB, and 556MB, respectively. The detailed system information that is extracted from the registry is described in the table. We note that our proposed framework can be applied into any Windows registry only if we extract the registry entries using the command above. Hence, we used four registries in the table as the examples to measure the performance of the proposed algorithms. We note that their sizes vary widely according to the systems where $Registry_1$ is the registry obtained immediately after the Windows system is installed. This implies that a lot of information is added and updated in the registry while the Windows operating system is running. In total, we collected 1.561 GB of Windows registry, where over 2 million registry keys are stored.

## 5.2 Experimental results

**5.2.1 Loading Windows registry into Hadoop distributed file system.** Fig 11 shows the processing time of the algorithm proposed for loading Windows registry into HDFS (See Section 4.3). Here, we measure the processing time varying the number of nodes and the number of CPUs.

As presented in Fig 11(a), when the number of CPUs is fixed to four, the processing time of the algorithm is reduced by up to approximately 1.73 times as the number of worker nodes

**Table 1. Characteristics of the collected Windows registry data.**

|  | *Registry₁* | *Registry₂* | *Registry₃* | *Registry₄* |
|---|---|---|---|---|
| Size | 240MB | 353MB | 413MB | 556MB |
| Number of entries | 374,266 | 523,477 | 629,524 | 794,723 |
| Installed date | 2020.02.25 | 2018.09.15 | 2020.03.13 | 2018.05.14 |
| CurrentBuildNumber | 18363 | 17763 | 17763 | 18363 |
| ProductName | Windows 10 Home | Windows 10 Enterprise | Windows 10 Enterprise | Windows 10 Home |

increases from one to eight. As presented in Fig 11(b), when the number of worker nodes is fixed to four, the algorithm processing time is reduced by up to approximately 2.94 times as the number of CPUs increases from one to eight. Finally, the processing time of the proposed algorithm, where eight worker nodes and eight CPUs are used, is reduced by up to approximately six times compared to the case where one worker node and one CPU are used. Owing to the overhead of the master node, the performance improvement is not exactly proportional to the number of worker nodes and CPUs. However, the results show a definite improvement in the processing performance of the proposed algorithms as the number of worker nodes and CPUs increases in the Apache Spark framework. The results, therefore, establish the scalability of the proposed algorithm.

**5.2.2 Scenario 1: Forensic for the target registry key.**   Fig 12 shows the processing time of the algorithm proposed for forensic for a target registry key (See Sec 4.4.1). Here, we increase the number of CPUs and the number of worker nodes. We used 'HKLM\SOFTWARE\Microsoft\Windows NT\CurrentVersion\ProfileList', which is associated with user profile information on the computer, as a target registry key. We measured the time for retrieving the registry entry for the target key from *Registry₄* 10 times and obtained its average time.

As presented in Fig 12(a), when the number of CPUs is fixed to four, the processing time is reduced by up to approximately 1.52 times as the number of worker nodes increases from one to eight. As presented in Fig 12(b), when the number of worker nodes is fixed to four, the processing time is reduced by up to approximately 2.35 times as the number of CPUs increases from one to eight. Here again, the results verify the scalability of the proposed algorithm.

**5.2.3 Scenario 2: Forensic on registry entries containing a target keyword.**   Fig 13 shows the processing time of the algorithm proposed for forensics on registry entries containing keywords (See Section 4.4.2). Here, we increase the number of CPUs and the number of worker nodes. We selected 10 keywords related to Windows registry forensics from top 30 keywords in 'Industrial Term', a list of keywords related to cyberattacks managed by Recorded Future

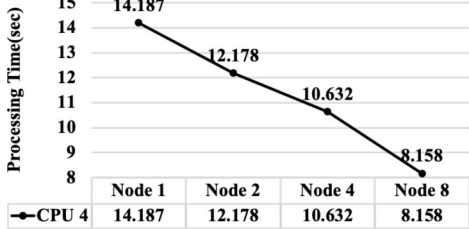

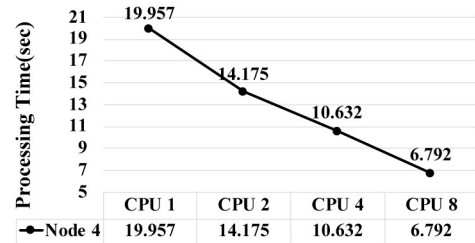

(**a**) The result with increasing the number of nodes.

(**b**) The result with increasing the number of CPUs.

**Fig 11. The processing time of the algorithm proposed for loading Windows registry into HDFS.** (**a**) The result with increasing the number of nodes. (**b**) The result with increasing the number of CPUs.

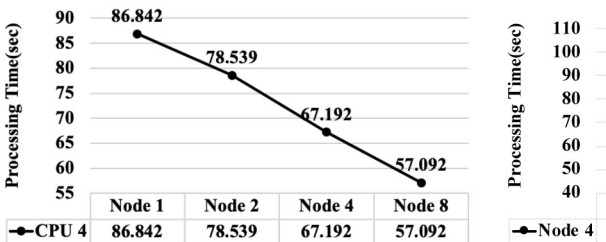
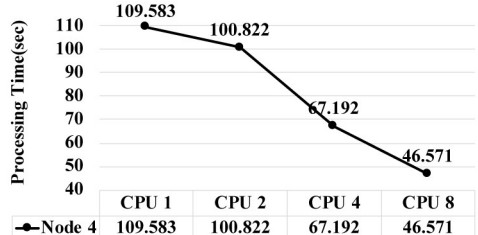

(a) The result with increasing the number of nodes.

(b) The result with increasing the number of CPUs.

**Fig 12. The processing time of the algorithm proposed for forensic for the target registry key.** (a) The result with increasing the number of nodes. (b) The result with increasing the number of CPUs.

(https://recordedfuture.com/). The selected 10 keywords are as follows: *'PoC', 'Exploit', 'RCE', 'CVE', 'Flash', 'Samba', 'PHP', 'RTF', 'OLE', 'CGI'*. We measured the time for searching registry entries containing each target keyword from $Registry_4$ and obtained its average time.

As presented in Fig 13(a), when the number of CPUs is fixed to four, the processing time of the algorithm is reduced by approximately 1.3 times as the number of worker nodes increases from one to eight. As presented in Fig 13(b), when the number of worker nodes is fixed to four, the processing time of the algorithm is reduced by up to approximately 1.91 times as the number of CPUs increases from one to eight. Here again, the results verify the efficiency and scalability of the proposed algorithm.

Fig 14 shows the actual results when the keywords *'PHP', 'Exploit'*, and *'Flash'* are given to $Registry_4$, respectively. As presented, the algorithm finds all the registry entries that contain a given keyword in the registry key path or the registry value. Fig 14(a) shows the case where a keyword, *'PHP'*, is included in registry key path; Fig 14(b) and 14(c) show the cases where keywords, *'Exploit', 'Flash'*, are included in the registry value, which is underlined.

**5.2.4 Scenario 3: Comparing the entire registry repositories.** Fig 15 shows the processing time of the proposed algorithm for comparing the entire two registry repositories (See Section 4.4.3). Here, we increase the number of CPUs and the number of worker nodes. We use $Registry_1$ and $Registry_4$ for the comparison.

As presented in Fig 15(a), when the number of CPUs is fixed to four, the processing time is reduced by up to approximately 1.31 times as the number of worker nodes increases from one to eight. As presented in Fig 15(b), when the number of worker nodes is fixed to four, the processing time is reduced by up to approximately 1.47 times as the number of CPUs increases

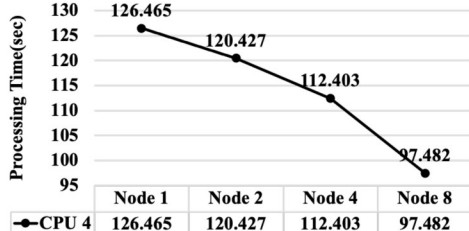
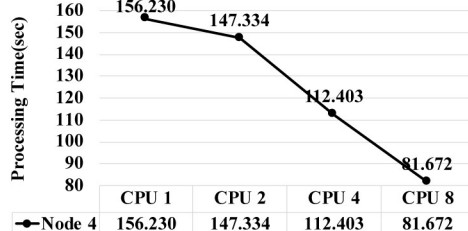

(a) The result with increasing the number of nodes.

(b) The result with increasing the number of CPUs.

**Fig 13. The processing time of the algorithm proposed for forensic on registry entries containing a target keyword.** (a) The result with increasing the number of nodes. (b) The result with increasing the number of CPUs.

> {HKLM : {SOFTWARE : {Classes : {.*php*3 : {PerceivedType : <u>text</u>}

(**a**) The given keyword is *'PHP'*.

> {HKLM : {SOFTWARE …{default : <u>*Exploit* Shield Brokeer</u>},
>
> {default : <u>*Exploit* Shield Class</u>},
>
> {default : <u>I*Exploit*Shield</u>}}}}}}

(**b**) The given keyword is *'Exploit'*.

> {HKLM : {SOFTWARE … {default : <u>*Flash*Broker</u>},
>
> {default : <u>Shockwave *Flash* Object</u>}}}

(**c**) The given keyword is *'Flash'*.

**Fig 14. The actual examples obtained by the algorithm for Scenario 2.** (**a**) The given keyword is '*PHP*'. (**b**) The given keyword is '*Exploit*'. (**c**) The given keyword is '*Flash*'.

from one to eight. Here once again, the results validate the efficiency and scalability of the proposed algorithm.

Table 2 shows the results of comparing the entire registry repositories when we compare $Registry_1$ with $Registry_2$ and $Registry_4$, respectively. The results are divided into two categories: 1) the registry key exists only in $RegRepo_1$, i.e., different registry keys, and 2) the same registry key exists in both registries, but their registry values are different, i.e., different registry values. When we compare $Registry_1$ with $Registry_4$, 243,321 different registry entries are identified, which occupy about 65.01% of $Registry_1$. Here, we indicate that the most cases correspond to the second category, i.e., different registry values. Specifically, the first category occupies only 0.71% while the second category occupies 99.28%. When we compare $Registry_1$ with $Registry_2$, 80,141 different registry entries are identified, which occupy about 21.41% of $Registry_1$. Here again, the most cases correspond to the second category.

Fig 16 shows the actual result of comparing between $Registry_1$ and $Registry_4$. Fig 16(a) shows an example where the presented registry key exists only in $Registry_1$. Fig 16(b) shows an example where the registry values, '*American Megatrends Inc.*' in $Registry_1$ and '*Phoenix Technologies Ltd.*' in $Registry_4$, are different for the same registry key, '{HKLM:{HARDWARE: {DESCRIPTION: {System:{BIOS:{BIOSVendoer:{}}}}}}}'.

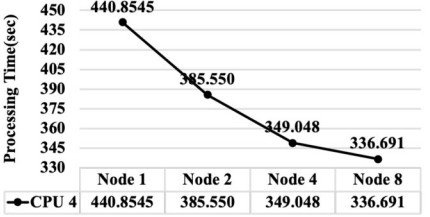

(**a**) The result with increasing the number of nodes.

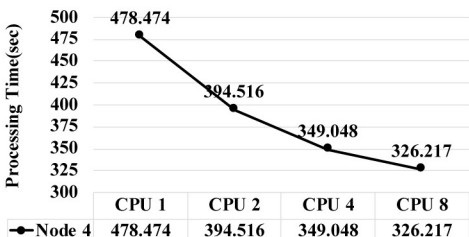

(**b**) The result with increasing the number of CPUs.

**Fig 15. The processing time of the presented algorithm for comparing the entire registry repositories in $Registry_1$ and $Registry_4$.** (**a**) The result with increasing the number of nodes. (**b**) The result with increasing the number of CPUs.

**Table 2. The result of comparing the entire registry repositories.**

| RegRepo₁ | RegRepo₂ | Different registry keys | Different registry values | Total differences |
|---|---|---|---|---|
| Registry₁ | Registry₂ | 1,638(2.04%) | 78,503(97.96%) | 80,141 |
| | Registry₄ | 1,739(0.71%) | 241,582(99.28%) | 243,321 |

**5.2.5 Comparing the performance of the proposed distributed algorithms with them in a single node.** Fig 17 compares the processing time of the proposed distributed algorithms on Apache Spark with them in a single node. We measure the processing time for the algorithms in a single node by configuring all the components for Apache Spark in the node, eliminating network overheads between nodes. For distributed algorithms, we present their results choosing the best environment setting in our distributed configuration according to the number of CPU cores and the number of worker nodes from the previous experiments, i.e., 8 worker nodes along with 4 CPU cores for each node and 4 worker nodes along with 8 CPU cores for each node. Here, we measure the processing time of a total of four algorithms, one for loading the Windows registry into HDFS in single system and the algorithms for three scenarios to perform forensic analysis on the Windows registry.

From the experimental results, we indicate that the proposed distributed algorithms outperform the results in a single node. Specifically, as shown in Fig 17(a), the processing time of loading the Windows registry into HDFS with the proposed distributed algorithm is reduced from 1.77. to 4.45 times compared to that in a single node. As presented in Fig 17(b), the processing time of forensics for the target registry key with the proposed distributed algorithm is reduced from 2.36 times to 2.9 times compared to that in a single system. As presented in Fig 17(c), the processing time of forensics on registry entries with the proposed distributed algorithm is reduced from 1.83 to 2.18 times compared to that in a single node. As presented in Fig 17(d), the processing time of comparing the entire registry entries with the proposed distributed algorithm is reduced from 1.59 times to 1.64 times compared to that in a single node. Because the distributed algorithms on Apache Spark require the inherent network and MapReduce overheads, this improvement of the processing performance verifies the efficiency and scalability of the proposed algorithms. Because we show that only eight nodes in a distributed environment are sufficient to show them, we can easily significantly improve its performance by adding more nodes.

> {HKU : {SOFTWARE … {ModuleInfo : {InstallDate : 2-25-2020},
> {SetupPath : D:\\SETUP\\}}}

(**a**) The registry key exists only in Registry₁.

> Registry₁ : {HKLM … {BIOS : {BIOSVendor : *American Megatrends Inc.*},
> {BIOSVersion : *2.02(D)*}}}}}}
> Registry₄ : {HKLM … {BIOS : {BIOSVendor : *Phoenix Technologies Ltd.*},
> {BIOSVersion : *W1ZD1210 X64*}}}}}}

(**b**) Different registry values for the same registry key.

**Fig 16. The examples obtained by the algorithm for Scenario 3.** (**a**) The registry key exists only in Registry₁. (**b**) Different registry values for the same registry key.

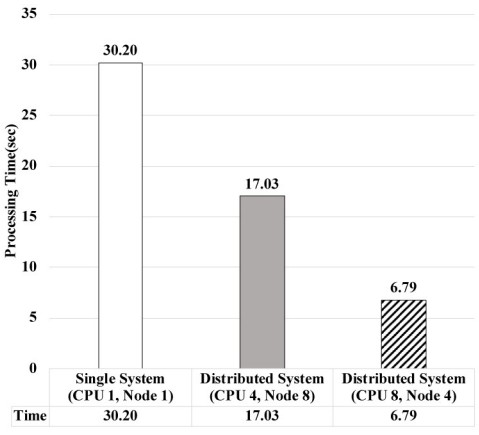

(**a**) Processing time of loading Windows registry into HDFS.

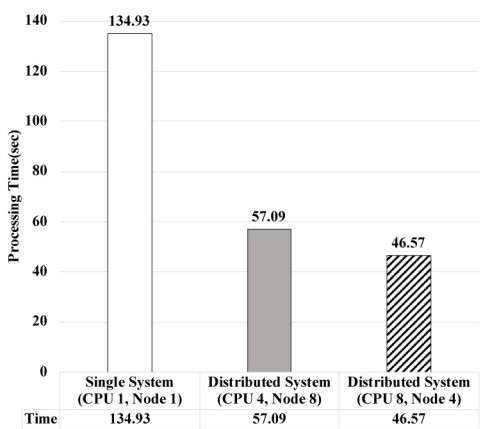

(**b**) Processing time of forensic for a target registry key.

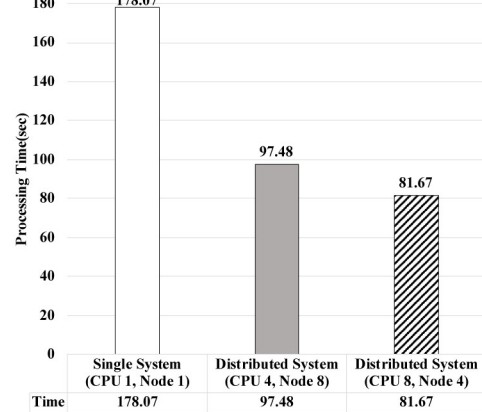

(**c**) Processing time of forensic on registry entries containing a target keyword.

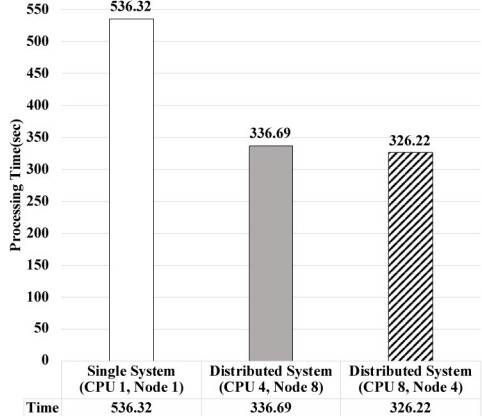

(**d**) Processing time of comparing the entire registry repositories.

**Fig 17. Comparing the processing time of the proposed distributed algorithms on Apache Spark with them in a single node.** (**a**) Processing time of loading Windows registry into HDFS. (**b**) Processing time of forensic for a target registry key. (**c**) Processing time of forensic on registry entries containing a target keyword. (**d**) Processing time of comparing the entire registry repositories.

## 6 Conclusions

In this study, we investigated large-scale digital forensic investigations on Apache Spark using a Windows registry. We devised distributed algorithms to analyze large-scale registry data collected from a number of Windows systems on the Apache Spark framework. First, we defined three main scenarios in which we classify the existing registry forensic studies into them. Second, we proposed an algorithm to load the Windows registry into HDFS for subsequent forensics. Third, we proposed a distributed algorithm for each defined forensic scenario using Apache Spark operations. Extensive forensic experiments performed on 1.52 GB of Windows registry data collected from four computers have shown that the proposed method can reduce the processing time by up to about 3.31 times as the number of CPUs was increased from 1 to 8 and the number of worker nodes from 2 to 8, validating the efficiency and scalability of the proposed algorithms.

Furthermore, we have shown the effectiveness of the framework to integrate and analyze large-scale Windows registries collected from multiple operating systems. For this purpose, we

managed them on a scalable distributed file system and proposed an efficient algorithm for extracting and transforming the Windows registry into HDFS. We have also shown the efficiency of the proposed forensic algorithms using Apache Spark operations as the parallelism of the cluster (i.e., the number of worker nodes and CPU cores) increases. Consequently, using the proposed method, we have shown that we can store and manage a large number of Windows registry repositories, which cannot be stored in a single or a small number of systems, on the proposed framework and can analyze them at a high speed. Additionally, we targeted the registry in Windows PCs to demonstrate the effectiveness of the proposed method. However, Windows registries are commonly used for every environment that supports the Windows operating system, including mobile devices and cloud services. This means that we can apply the proposed approach to perform forensic analysis on registries obtained from different environments.

## Author Contributions

**Conceptualization:** Jun-Ha Lee, Hyuk-Yoon Kwon.

**Data curation:** Jun-Ha Lee.

**Formal analysis:** Jun-Ha Lee, Hyuk-Yoon Kwon.

**Funding acquisition:** Hyuk-Yoon Kwon.

**Investigation:** Jun-Ha Lee, Hyuk-Yoon Kwon.

**Methodology:** Jun-Ha Lee, Hyuk-Yoon Kwon.

**Project administration:** Hyuk-Yoon Kwon.

**Software:** Jun-Ha Lee.

**Supervision:** Hyuk-Yoon Kwon.

**Validation:** Hyuk-Yoon Kwon.

**Writing – original draft:** Jun-Ha Lee, Hyuk-Yoon Kwon.

**Writing – review & editing:** Jun-Ha Lee, Hyuk-Yoon Kwon.

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
