## [Decision Letter · Decision Letter 0]

15 Dec 2021

PONE-D-21-30776Large-Scale Digital Forensic Investigation for Windows Registry on Apache SparkPLOS ONE

Dear Dr. Kwon,

Thank you for submitting your manuscript to PLOS ONE. After careful consideration, we feel that it has merit but does not fully meet PLOS ONE’s publication criteria as it currently stands. Therefore, we invite you to submit a revised version of the manuscript that addresses the points raised during the review process.

We look forward to receiving your revised manuscript.

Kind regards,

Qingzhong Liu, PhD

Academic Editor

PLOS ONE

Journal Requirements:

"This work was supported in part by the National Research Foundation of Korea(NRF) grant funded by the Korea government(MSIT) (No.2021R1F1A1064050), and in part by the Basic Science Research Program through the National Research Foundation of Korea(NRF) funded by the Ministry of Education (No.

547 2019R1A6A1A03032119)"

Reviewers' comments:

Reviewer's Responses to Questions

**Comments to the Author**

1. Is the manuscript technically sound, and do the data support the conclusions?

Reviewer #1: Partly

2. Has the statistical analysis been performed appropriately and rigorously? 

Reviewer #1: Yes

3. Have the authors made all data underlying the findings in their manuscript fully available?

Reviewer #1: Yes

4. Is the manuscript presented in an intelligible fashion and written in standard English?

Reviewer #1: Yes

5. Review Comments to the Author

Reviewer #1: In the research paper “Large-Scale Digital Forensic Investigation for Windows Registry on Apache Spark” the authors developed a technique to forensically analyze a Windows registry using Apache Spark. The authors developed algorithms to parse the data and use it with Apache Spark. They also evaluated their proposed system’s performance and compared it with previous approaches. The research is interesting and relevant because it explores the use of novel technologies to decrease processing time in forensic investigations.

The introduction provides some initial information on the topic along with the contributions of the paper. The literature review and background section provide good information on relevant research and relevant information. The methodology describes the experimental setup along with the proposed algorithms and their implementation. The methodology section does a great job explaining the algorithms along with providing examples of how the algorithms work. The results explain in detail the advantages of their proposed approach along with an investigation of the different configurations. Overall, the paper is interesting and relevant. However, the paper needs some modifications before publishing.

The paper needs some citations:

In the second paragraph of the introduction, the authors describe the information stored in the Windows registry. The authors should provide a citation for this information.

In section 2.1 the authors mention RegEdit but do not cite it. They should cite it

In section 2.1, the authors mention that RegEdit can extract data into a text file and briefly describe its structure. The authors should cite this information.

The authors mention through the paper “Apache Spark”, “HDFS”, and “RegEdit” but do not cite their occurrences. They should.

The authors should cite “Google Cloud Platform”, “Apache Hadoop 2.10.0”, and “Apache Spark 2.4.7” in section 5.1

The Introduction does not properly convey the importance of the paper. The authors mention that “The existing approach analyzes Windows registry targeting on a single Windows system. On the other hand, in our framework, we extract the Windows registry from several Windows systems and transform and load them into the Hadoop distributed file system (HDFS) on a Hadoop cluster“. However, they do not mention the flaws of single system analysis or of existing analysis nor why it would be necessary or important to process registry information from several Windows systems. The authors should modify the introduction by explicitly stating the shortcomings of single system analysis and the advantages of using HDFS.

In the Related Work section the authors summarize existing relevant works on Windows Forensics, Big Data Forensics, and Digital Forensics using Apache Spark and MapReduce. The authors only mention how their research is different from existing approaches in section 3.2 at the start of the paragraph. This statement should be moved to the end of the section to convey to the reader that their work is different than the existing approaches. Similar for section 3.3 and 3.1. The authors should write a paragraph at the end of those sections illustrating how their research is different from the other literature in those sections.

In section 5.1 the authors state “we measure the performance of the algorithm for loading registry data into HDFS proposed in Section 4.3 using up to eight nodes”. How are they measuring the performance? What metrics are being used? The authors should state this information in this section for experiment replication purposes.

In section 5.1 in the second paragraph, the authors state “We collected actual Windows registry data from four different systems running Windows operating systems”. The authors should define what the specifications of these “Windows operating systems” are along with what data these systems were running for experiment replication purposes.

In section 5.2.1 the authors state “Here, we measure the performance varying the number of

nodes and the number of CPUs.”. How is the performance being measured in this section? The authors should state the units and how the data was acquired for experiment replication purposes.

6. PLOS authors have the option to publish the peer review history of their article (what does this mean?). If published, this will include your full peer review and any attached files.

Reviewer #1: No

---

## [Author Response · Author response to Decision Letter 0]

4 Feb 2022

We deeply appreciate the reviewer’s careful concerns. Please refer to the attached a file labeled "response to reviewers".

---

## [Decision Letter · Decision Letter 1]

8 Apr 2022

Large-Scale Digital Forensic Investigation for Windows Registry on Apache Spark

PONE-D-21-30776R1

Dear Dr. Kwon,

We’re pleased to inform you that your manuscript has been judged scientifically suitable for publication and will be formally accepted for publication once it meets all outstanding technical requirements.

Kind regards,

Qingzhong Liu, PhD

Academic Editor

PLOS ONE

Additional Editor Comments (optional):

Reviewers' comments:

Reviewer's Responses to Questions

**Comments to the Author**

1. If the authors have adequately addressed your comments raised in a previous round of review and you feel that this manuscript is now acceptable for publication, you may indicate that here to bypass the “Comments to the Author” section, enter your conflict of interest statement in the “Confidential to Editor” section, and submit your "Accept" recommendation.

Reviewer #1: All comments have been addressed

2. Is the manuscript technically sound, and do the data support the conclusions?

Reviewer #1: (No Response)

3. Has the statistical analysis been performed appropriately and rigorously? 

Reviewer #1: (No Response)

4. Have the authors made all data underlying the findings in their manuscript fully available?

Reviewer #1: (No Response)

5. Is the manuscript presented in an intelligible fashion and written in standard English?

Reviewer #1: (No Response)

6. Review Comments to the Author

Reviewer #1: (No Response)

7. PLOS authors have the option to publish the peer review history of their article (what does this mean?). If published, this will include your full peer review and any attached files.

Reviewer #1: No

---

## [Editor Report · Acceptance letter]

12 Apr 2022

PONE-D-21-30776R1 

Large-Scale Digital Forensic Investigation for Windows Registry on Apache Spark 

Dear Dr. Kwon:

I'm pleased to inform you that your manuscript has been deemed suitable for publication in PLOS ONE. Congratulations! Your manuscript is now with our production department. 

Kind regards, 

on behalf of

Dr. Qingzhong Liu 

Academic Editor

PLOS ONE